# Equilibrium CT Texture Analysis for the Evaluation of Hepatic Fibrosis: Preliminary Evaluation against Histopathology and Extracellular Volume Fraction

**DOI:** 10.3390/jpm10020046

**Published:** 2020-05-29

**Authors:** Jason Yeung, Balaji Ganeshan, Raymond Endozo, Andrew Hall, Simon Wan, Ashley Groves, Stuart A. Taylor, Steve Bandula

**Affiliations:** 1Centre for Medical Imaging, University College London, 250 Euston Road, London NW1 2PG, UK; stuart.taylor1@nhs.net (S.A.T.); s.bandula@ucl.ac.uk (S.B.); 2Institute of Nuclear Medicine, University College London, 235 Euston Road, London NW1 2BU, UK; b.ganeshan@ucl.ac.uk (B.G.); raymond.endozo@nhs.net (R.E.); mwan@nhs.net (S.W.); a.groves@ucl.ac.uk (A.G.); 3Department of Pathology, Royal Free Hospital, Pond Street, London NW3 2QG, UK; andrewhall1@nhs.net

**Keywords:** liver, CT, cirrhosis, texture, analysis, equilibrium

## Abstract

Background: Evaluate equilibrium contrast-enhanced CT (EQ-CT) texture analysis (EQ-CTTA) against histologically-quantified fibrosis, serum-based enhanced liver fibrosis panel (ELF) and imaging-based extracellular volume fraction (ECV) in chronic hepatitis. Methods: This study was a re-analysis of image data from a previous prospective study. Pre- and equilibrium-phase post-IV contrast CT datasets were collected from patients with chronic hepatitis with contemporaneous liver biopsy and serum ELF measurement between April 2011 and July 2013. Biopsy samples were analysed to derive collagen proportionate area (CPA). EQ-CTTA was performed with a filtration histogram technique using texture analysis software, with texture quantification using statistical and histogram-based metrics (mean, skewness, standard deviation, entropy, etc.). Association between pre-contrast and EQ-CTTA against CPA, ECV and ELF was evaluated using Spearman’s rank correlation coefficient (r_s_). Results: Complete datasets collected in 29 patients (16 male; 13 female), mean age (range): 49 (22–66 years). Liver ECV, CPA and ELF had a median (interquartile range) of 0.26 (0.24–0.29); 5.0 (3.0–13.7) and 9.71 (8.39–10.92). Difference in segment VII hepatic CTTA (medium texture scale) between EQ-CT and pre-contrast images was significantly and positively associated with ELF score (mean: r_s_ = 0.69, *p* < 0.001; skewness: r_s_ = 0.57, *p* = 0.007). Significant negative associations were observed between pre-contrast and EQ-CT whole hepatic CTTA (coarse texture scale) with CPA (pre-contrast, SD: r_s_ = −0.66, *p* < 0.001) and ECV (EQ-CT, entropy: r_s_ = −0.58, *p* = 0.006). Conclusions: Hepatic EQ-CTTA demonstrates significant association with validated markers of liver fibrosis, suggesting a role in non-invasive quantification of severity in diffuse fibrosis.

## 1. Introduction

Chronic liver disease is a major global public health issue. Viral hepatitis [1] resulted in an estimated 1.29 million deaths worldwide in 2010 [2], and in the United States of America, non-alcoholic steatohepatitis (NASH) has become the most rapidly growing aetiology for acute-on-chronic liver failure related hospitalization and disease burden [3]. 

Hepatocellular injury leads to an inflammatory reaction which results in diffuse deposition of fibrous tissue within the parenchyma, leading to eventual cirrhosis. This is mediated by hepatic stellate cells, which, in health, are non-proliferative and quiescent. When activated into a myofibroblast-like phenotype, they lay down collagenous extracellular matrix (ECM), leading to expansion of the extracellular space, distortion of normal architecture, abnormal blood flow and eventual impairment of hepatic function [4]. Cirrhosis is defined pathologically by fibrosis and the subsequent development of nodules of regenerating hepatocytes. It can be classified into progressive stages according to the degree of fibrosis and the type of nodules: incomplete septal fibrosis (incomplete bridging fibrosis with no nodules), early fibrosis (thin bridging fibrosis with dissecting nodules), moderately advanced fibrosis (thick bridging fibrosis with dissecting nodules) and advanced fibrosis or cirrhosis (thick bridging fibrosis with regenerative hyperplastic nodules) [5].

Assessment of fibrosis plays a central role in diagnosis, staging and assessing response to treatment. Liver biopsy is the reference standard test for assessment of diffuse hepatic fibrosis. However, it is an invasive procedure with associated risk of complications [6] and is prone to sampling error [7]. Non-invasive techniques take advantage of altered tissue mechanical properties, (e.g., tissue elastography) or evaluate products of matrix synthesis/degradations and the enzymes involved (e.g., serum-based enhanced liver fibrosis panel [ELF]) [8].

Equilibrium imaging is an emergent technique which uses extracellular contrast to evaluate the fractional tissue extracellular volume (ECV), which is expanded in fibrosis and has shown association with biopsy-derived collagen-proportionate area (CPA) and ELF [8,9]. CT texture analysis (CTTA) is a method of quantifying heterogeneity of a region by analysing the distribution and/or the relationship of pixel or voxel grey levels in the image. By probing patterns and heterogeneity information within the imaging data, microstructure tissue elements at various scales are evaluated. Studies have shown potential for CTTA in detecting and discriminating between stages of hepatic fibrosis [10,11,12] and also in NASH [13] and non-alcoholic fatty liver disease [14] using standard diagnostic CECT. As equilibrium contrast-enhanced CT (EQ-CT) imaging enhances the expanded ECM, we hypothesise that textural analysis of this imaging data could provide new insights, picking out specific patterns of parenchymal histopathological change and potentially provide a more sensitive evaluation of fibrosis.

Our study proposes to evaluate CTTA in pre-contrast and hepatic equilibrium imaging against histologically-quantified fibrosis (CPA), serum-based ELF as well as imaging-based ECV in chronic hepatitis.

## 2. Materials and Methods

This study was a re-analysis of prospectively collected data and was approved by the regional ethics committee (reference number 09/H0716/75), with all patients giving written consent. 

### 2.1. Patient Selection

Pre-contrast CT datasets were available from 29 patients with chronic hepatitis who had undergone segment VII liver biopsy and serum sampling for ELF measurement in a previous prospective study validating CT-derived ECV as a biomarker of liver fibrosis [9]. Post-IV contrast image datasets were available from 21 of these patients. Patients were recruited between April 2011 and July 2013 from a chronic liver disease outpatient service at a single tertiary referral centre (Royal Free University Hospital, Hampstead, England). Inclusion criteria were serologically confirmed hepatitis B or C and clinically indicated liver biopsy. Exclusion criteria were contraindication to contrast agent (iodinated contrast allergy or eGFR rate <60 mL/min), HIV+, previous liver transplantation or a background of significant alcohol intake (men: >32 g/day; women: >24 g/day). In total, 190 patients met the inclusion criteria. In total, 150 were excluded (32: HIV+; 22: liver transplantation; 31: contraindication to contrast; 64: significant alcohol consumption). Please refer to Figure 1 for a flow diagram of the participants through the original study. 

### 2.2. Histological Analysis

Liver biopsy was performed either percutaneously with a Menghini technique [15] using a 16-gauge needle from the right lobe or via the transjugular approach with up to four passes of a 19-gauge needle through either the right or middle hepatic vein [16]. Only biopsy samples of ≥12 mm were included for analysis to ensure reliable histopathological evaluation. Tissues samples were formalin-fixed, paraffin-embedded and stained with haematoxylin and eosin and Gordon and Sweet stain for reticulin. An experienced hepatopathologist recorded the stage of fibrosis as per the Ishak system [17]. A further tissue section was stained with picro-Sirius red for collagen-proportionate area (CPA) quantification by digital image analysis [18], performed by another operator (A.H.) independent of fibrosis staging using a digital camera on the macro setting for low-magnification whole-section image capture. Image artefacts and structural collagen in portal tracts and vascular walls were removed and analysis was performed using software (Zeiss KS300; Carl Zeiss Microscopy, Jena, Germany).

### 2.3. EQ-CT

Iodinated contrast is an extravascular and extracellular tracer. After intravenous delivery and sufficient time for equilibration, it partitions in water between cells in blood (plasma) and tissue (interstitial fluid). Tissue interstitial volume is increased in fibrosis. Ratio of signal change before and after contrast in blood and tissue is the partition coefficient (ratio of volumes of distributions). If the plasma volume (1—hematocrit) is substituted, then tissue extracellular volume (ECV) can be calculated [19].

EQ-CT was performed with a 64-detector row CT scanner (Somatom Sensation 64; Siemens Medical Solutions, Erlangen, Germany) no more than two weeks prior to liver biopsy. Two 5 mL blood samples were collected before CT scan to derive blood haematocrit and for ELF analysis.

A pre-contrast scan was acquired through the upper abdomen to include the whole liver with the following parameters: effective tube current–time product, 250 mAs; pitch, 0.8; tube voltage, 120 kV; section collimation, 24 mm × 1.2 mm; gantry rotation time, 500 msec; reconstructed to a section thickness of 10 mm with a soft-tissue convolution kernel (B20f). Iodinated contrast agent iohexol (Omnipaque 300; GE Healthcare, Cork, Ireland) was then administered as a bolus dose of 1 mL/kg at 3 mL/sec; followed immediately by an infusion of 1.88 mL/kg per hour to achieve contrast agent steady state [20]. Scan volume was repeated after 30 min of contrast infusion.

Image analysis was performed with software (OsiriX v4.1.2; OsiriX Foundation, Geneva, Switzerland) by a radiologist of eight years of experience (S.B.) who was blinded to biopsy findings. On three contiguous sections reconstructed to a thickness of 10 mm, a peripherally based wedge-shaped region of interest (ROI) was drawn in segment VII of the liver (mean area, 34.5 cm^2^) to include the greatest area of liver parenchyma while avoiding visible vessels or incidental hepatic lesions. Imaging biopsy matching was at the level of the right lobe only. Another elliptical ROI was drawn on the same section within the abdominal aorta (mean area, 1.6 cm^2^) taking care to avoid the aortic wall and any atheromatous plaque. These ROIs that were drawn on the post-contrast images were then copied onto the matching sections of the pre-contrast images (Figure 2). Mean attenuation (in Hounsfield units) for liver parenchyma and blood were measured and the liver ECV was calculated using the following formula:ECV_liver_ = (1 − haematocrit) × (δHU_liver_/δHU_blood_)
where δHU is the change in HU between equilibrium phase and pre-contrast. 

### 2.4. CTTA

CTTA was performed by a single trained reader under the supervision of two experienced abdominal radiologists (12 and 22 years of experience) using a commercially available texture analysis research software platform (TexRAD provided by Feedback Medical Ltd., Cambridge, UK) using the pre-contrast and equilibrium contrast CT datasets. Representative single-slice images at the level of the hepatic porta were used, as it demonstrated the largest cross-sectional area of the liver [6]. A single ROI was drawn manually to include the whole liver but excluding the major vessels to obtain a total liver measurement (Figure 3). A second ROI was drawn around segment VII (Figure 4). CTTA comprised a filtration histogram technique, where the filtration step (using a Laplacian of Gaussian band-pass spatial filter similar to a non-orthogonal Wavelet) selectively extracted and enhanced features of different sizes and intensity variation based on a particular spatial scale filter (SSF) value corresponding to fine (SSF = 2 mm), medium (SSF = 3–5 mm) and coarse (SSF = 6 mm) texture scales. Figure 3 and Figure 4 highlight the filtration step which produces the derived filtered images/maps. The filtration step was followed by quantification of textures using statistical and histogram-based parameters which included mean grey-level intensity (mean), standard deviation (SD), entropy (irregularity), mean of the positive pixels (MPPs), skewness (asymmetry) and kurtosis (peakedness or sharpness) at each SSF value. Texture quantification without filtration (SSF = 0) was also quantified and acted as a control. A simulation study by Miles et al. [21] described filtration histogram-based CTTA in detail, demonstrating how this technique reflects different components of heterogeneity (object size, number of objects and variation in density of these objects in relation to the background) and its association with visual perception of the image features. 

### 2.5. ELF Test Biomarker

Serum was separated from blood samples collected prior to CT examination. This was stored at −80 °C before being transported to an independent laboratory (iQur Limited, London, England). Proprietary assays (Siemens Healthcare Diagnostics, Tarrytown, NY, USA) were used in conjunction with an automated analyser (Advia Centaur; Siemens Healthcare Diagnostics) in accordance with manufacturer instructions to measure levels of tissue inhibitor matrix metalloproteinase 1, hyaluronic acid and amino-terminal propeptide of type-III collagen [9]. Results were combined using an algorithm to generate a fibrosis score [8]. 

### 2.6. Statistical Analysis

All statistical tests were performed using SPSS 22 (IBM Corp. Released 2013. IBM SPSS Statistics for Macintosh, Version 22.0. Armonk, NY: IBM Corp). Spearman’s rank correlation was used to evaluate association between texture analysis parameters of pre-contrast, equilibrium-contrast and the difference between the two (EQ-CT minus pre-contrast) of whole liver and segment VII against CPA, ECV, ELF and Ishak fibrosis score. *p* ≤ 0.05 was considered to be significant. 

## 3. Results

Pre-contrast image data, ELF and CPA datasets were available in 29 patients (16 male; 13 female); average age (range): 49 (22–66) years). Liver ECV, CPA and ELF had a median (interquartile range) of 0.26 (0.24–0.29); 5.0 (3.0–13.7) and 9.71 (8.39–10.92). EQ-contrast image data were available in 21 of these patients.

The mean and the skewness of the difference in segment VII hepatic CTTA at medium to coarse texture scales between EQ-CT and pre-contrast images were significantly and positively associated with ELF score (best at medium texture scale—mean: r_s_ = 0.69, *p* < 0.001; skewness: r_s_ = 0.57, *p* = 0.007). Please refer to Table 1 and Figure 5 and Figure 6. However, this association was not demonstrated in whole liver analysis.

There were significant negative associations between both pre-contrast and EQ-CT whole hepatic CTTA at medium to coarse texture scale and CPA and ECV, where the negative correlations were higher and more significant at pre-contrast compared to EQ-CT whole hepatic CTTA, particularly for CPA (best at coarse texture scale—pre-contrast SD: r_s_ = −0.66, *p* < 0.001; pre-contrast entropy: r_s_ = −0.60, *p* = 0.001; pre-contrast MPP: r_s_ = −0.55, *p* = 0.002; EQ-CT SD: r_s_ = −0.52, *p* = 0.015; EQ-CT MPP: r_s_ = −0.58, *p* = 0.005) and ECV (best at coarse texture scale—pre-contrast SD: r_s_ = −0.51, *p* = 0.004; pre-contrast entropy: r_s_ = −0.46, *p* = 0.011; pre-contrast MPP r_s_ = −0.46, *p* = 0.013; EQ-CT entropy: r_s_ = −0.58, *p* = 0.006; EQ-CT SD: r_s_ = −0.56, *p* = 0.008; EQ-CT MPP: r_s_ = −0.52, *p* = 0.016). 

The above significant associations appear mainly preserved within medium to coarse texture-scale analysis. However, at more finer textures, the associations weaken and are not significant. Please refer to Table 2 and Figure 7 and Figure 8. 

## 4. Discussion

Previous work by Lubner et al. has demonstrated that textural features within liver parenchymal on standard CT scanning can identify underlying hepatic fibrosis [10]. In this study, we develop this concept further, using equilibrium-phase CT imaging to focus on changes within the extracellular matrix, and demonstrate association between textural features and hepatic extracellular volume fraction, histological collagen proportionate area (CPA), and the plasma enhanced liver fibrosis (ELF) biomarker.

EQ (equilibrium) subtracted enhancement texture analysis (EQ-CT minus pre-contrast CTTA) of segment VII, in particular mean and skewness at medium texture scale, was significantly and positively associated with ELF score. Miles et al. [21] proposed a framework for interpreting filtration histogram-based CTTA (employed in this study), where increased mean intensity value reflects increased object brightness and increased skewness reflects preponderance of bright features. Subtraction of pre-contrast from EQ data specifically aims to highlight contrast within the hepatic interstitum. This association agrees with our hypothesis that the bridging and nodule pattern of fibrosis, even in early stages of fibrosis, would be reflected in greater medium and coarse texture signal. Increased mean and increased skewness reflect a greater number and spread of medium–coarse features within the hepatic interstitum with increasing fibrosis, in contrast to the homogenous texture of healthy liver. 

This association was not demonstrated in whole liver analysis. Tissue biopsies were obtained from segment VII and this finding likely reflects differing extent of disease in other parts of the liver (fibrosis within the liver is often heterogeneous with areas or segments being spared). 

An alternative explanation could be that as the hepatic interstitum only reflects a small component of a fundamental hepatic lobule, whole liver analysis may dilute the effect of diseased foci (i.e., increasing the confounding effects of the hepatocellular tissue components outlined below), thus reducing the impact of a texture signal within the proportionally small volume interstitum.

Whole liver and segment VII SD, entropy and MPP showed strong negative correlations with both CPA and ECV, where the negative correlations were higher and more significant for pre-contrast compared to EQ-contrast texture analysis at medium and coarse texture scales. Miles et al. [21] explained lower SD as reflecting fewer number of features, lower entropy as lesser irregularity and lower MPP as fewer bright features. This correlation is the inverse of expected, possibly due to the fewer clusters of pixels (SD), with lower density texture features (entropy and MPP) at medium to coarser texture scales within the adjacent liver parenchyma (i.e., volume outside the interstitum which does not take up contrast) with increasing fibrosis (CPA and ECV). This may reflect background loss of hepatocytes in advancing fibrosis. This could also explain why the negative correlation is more significant with pre-contrast than with EQ-contrast CTTA. The effects of contrast within the interstitum in EQ-contrast CTTA, which will highlight interstitial fibrotic change and pick up more bright coarse features, may be offsetting the above effect from background loss of hepatocytes.

Another factor for the negative association could be that the CPA methodology used focuses on quantifying larger scale fibrosis (e.g., portal tract expansion, cirrhotic nodules, etc.). As the images are taken at a lower resolution than those traditionally seen down a brightfield microscope, peri-sinusoidal fibrosis can be poorly or inaccurately quantified. CTTA may be more reflective of parenchymal collagen deposition rather than larger structural fibrosis.

This study had a number of limitations. Texture analysis was only performed on a single slice of the liver CT rather than the entire volume. Some reported studies use three-dimensional ROIs for texture analysis, particularly for assessment of neoplasms [22,23,24]. Other studies have shown that the use of a single slice is sufficient for sampling and extracting subtle features in hepatic fibrosis [10] because it is a diffuse process in a relatively large organ and will therefore demonstrate less location-dependent variance when compared to tumours, which are inherently heterogeneous.

Despite the care taken to avoid visible hepatic vessels when drawing the ROIs, it is possible that some portion of a vessel may be inadvertently included in the analysis due to the difficulty in identifying the vessels in a fibrotic liver, thereby introducing error into the subtracted data.

Our study sample population was also limited to patients with chronic hepatitis B or C. This prevented meaningful comparison with data from the Lubner et al. study [10], which included patients with other chronic liver diseases including alcohol-related liver disease, non-alcoholic fatty liver disease and primary sclerosing cholangitis. Further larger cohort studies, including patients without liver disease, should be undertaken to confirm the suitability of CTTA in assessing and quantifying liver fibrosis in this cohort.

The filtration histogram-based CTTA used in our study is a widely used methodology which has undergone a qualification/validation process [10,11]. The approach comprises a limited number of validated parameters rather that the full set of CTTA parameters described in the literature. Use of the full parameter set (several hundred) would increase the rate of significant correlations just by chance.

In this study, we did not use a statistical correction (e.g., Bonferroni correction) for the multiple comparisons performed, as this was likely to be too conservative for the filtration histogram technique employed. The filtration histogram technique comprises multiple filter scales which are related (not independent) to each other and therefore Bonferroni correction (which assumes each test as independent) is therefore not suitable. Furthermore, the associations observed in our study between texture metrics and clinical markers were found at several separate filter scales, again suggesting significance.

CT plays an important diagnostic role in patients with chronic liver disease (e.g., in screening for hepatocellular carcinoma and signs of portal hypertension) and texture analysis could be easily integrated into existing workflows. Widespread availability and time effectiveness of CT imaging also confers benefits (e.g., cost and efficiency) when compared to other imaging techniques such as MRI. Compared to US-based techniques for evaluation of fibrosis, it may be less operator dependent and more reproducible.

The ability to non-invasively recognise and quantify fibrosis in at-risk populations could be a useful adjunct, alleviating the need for repeated invasive biopsies. Future studies will need to validate these results in larger populations and compare with other quantitative imaging biomarkers including elastography.

## 5. Conclusions

Hepatic equilibrium-phase CT texture (EQ-CTTA) analysis demonstrates significant associations with validated markers of liver fibrosis, namely biopsy-derived collagen-proportionate area, serum enhanced liver fibrosis panel and imaging-based extracellular volume. Our small proof-of-concept study indicates that EQ-CTTA can help quantify diffuse fibrosis in chronic liver disease. These results should be validated in a larger group of patients in a prospective study.

## Figures and Tables

**Figure 1 jpm-10-00046-f001:**
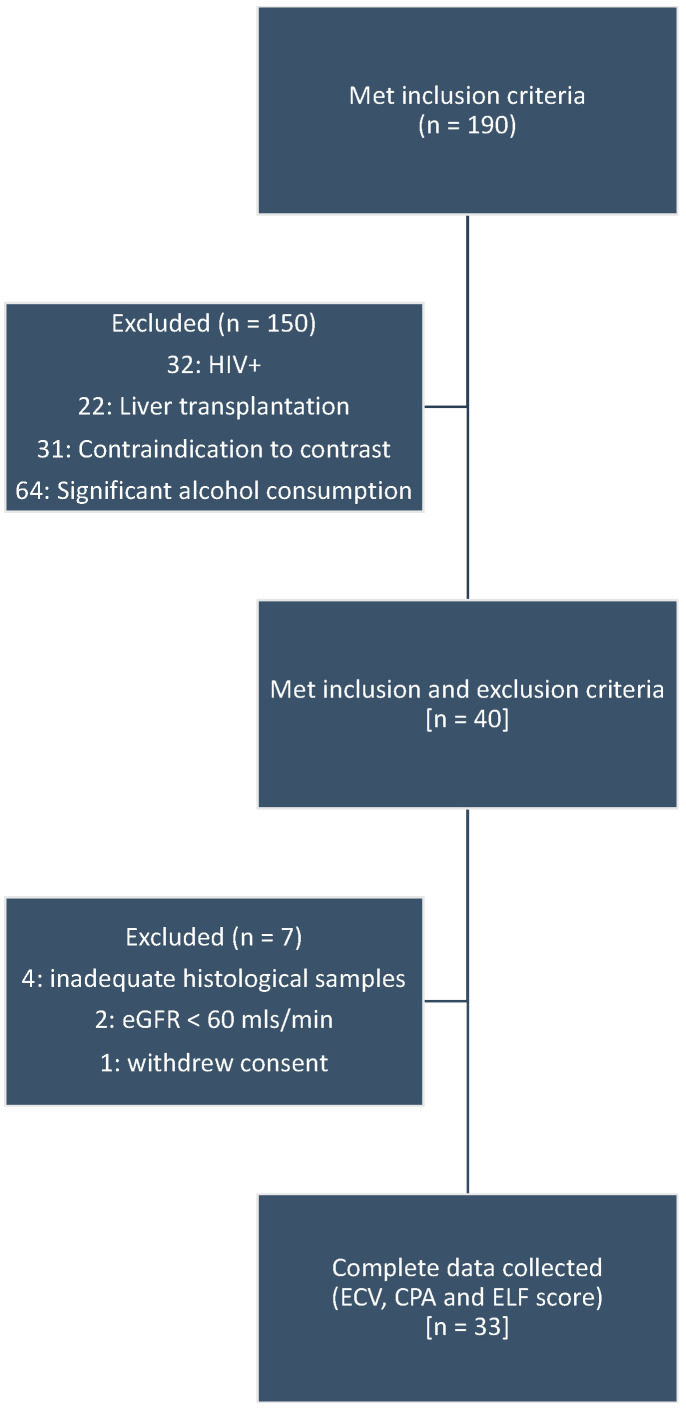
Flow diagram of participants through the original study, from which the datasets were used for re-analysis in this study.

**Figure 2 jpm-10-00046-f002:**
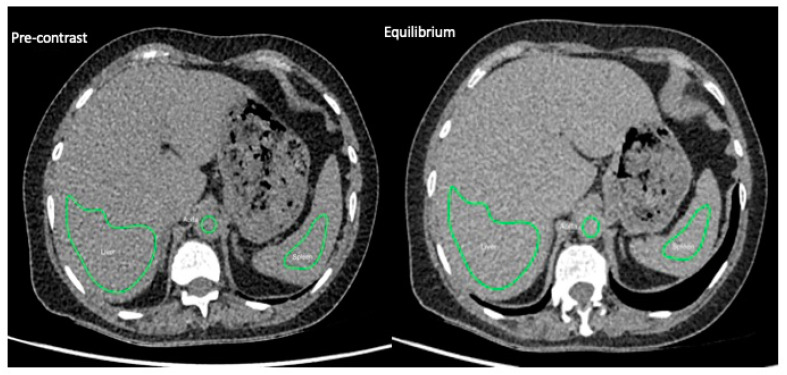
EQ-CT images pre- and at EQ-contrast phase through the upper abdomen, showing example regions of interest drawn in the liver, spleen, and abdominal aorta. (EQ-CT: equilibrium contrast-enhanced CT; EQ: equilibrium.)

**Figure 3 jpm-10-00046-f003:**
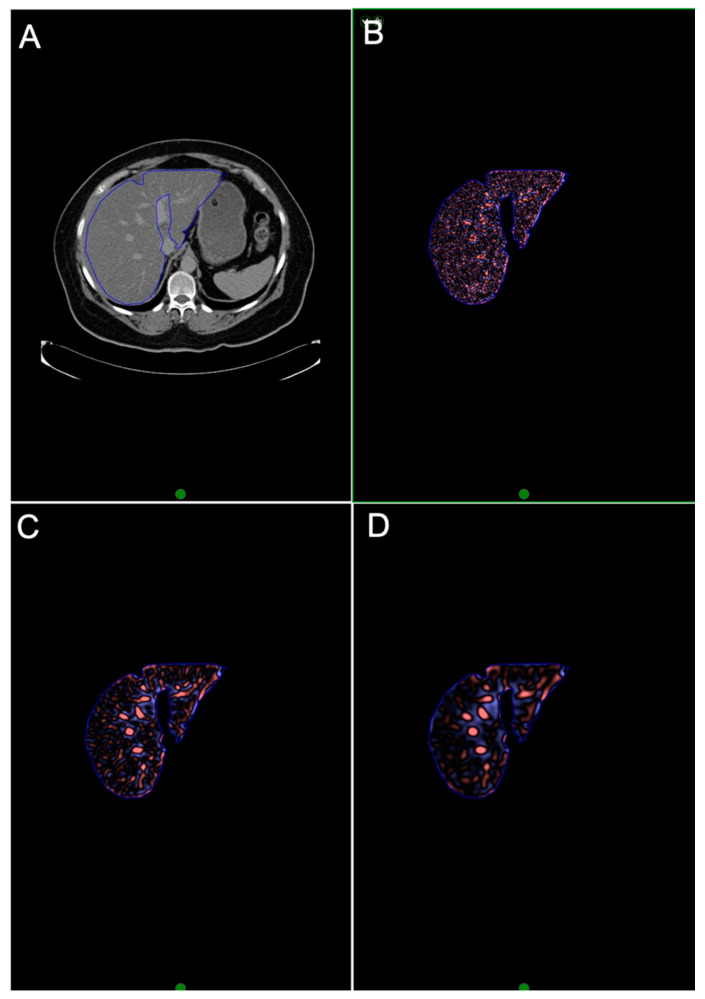
Texture analysis of EQ-CT data in whole liver. (**A**) EQ-CT (post-contrast phase) showing whole liver region of interest. (**B**–**D**) highlights the initial filtration step, where **B** reflects fine (features of 2 mm in size) texture map, **C** reflects medium (4 mm) texture map and **D** reflects coarse (6 mm) texture map. (EQ-CT: equilibrium contrast-enhanced CT.)

**Figure 4 jpm-10-00046-f004:**
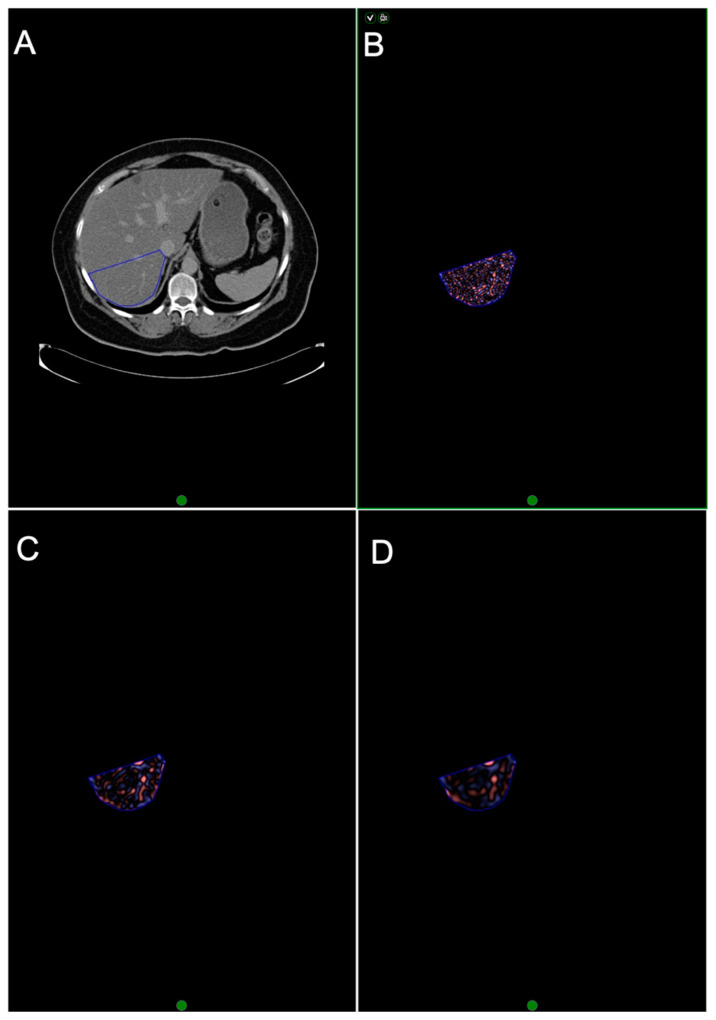
Texture analysis of EQ-CT data in liver segment VII. (**A**) EQ-CT (post-contrast phase) showing liver segment VII region of interest. (**B**–**D**) highlights the initial filtration step, where **B** reflects fine (features of 2 mm in size) texture map, **C** reflects medium (4 mm) texture map and **D** reflects coarse (6 mm) texture map. (EQ-CT: equilibrium contrast-enhanced CT.)

**Figure 5 jpm-10-00046-f005:**
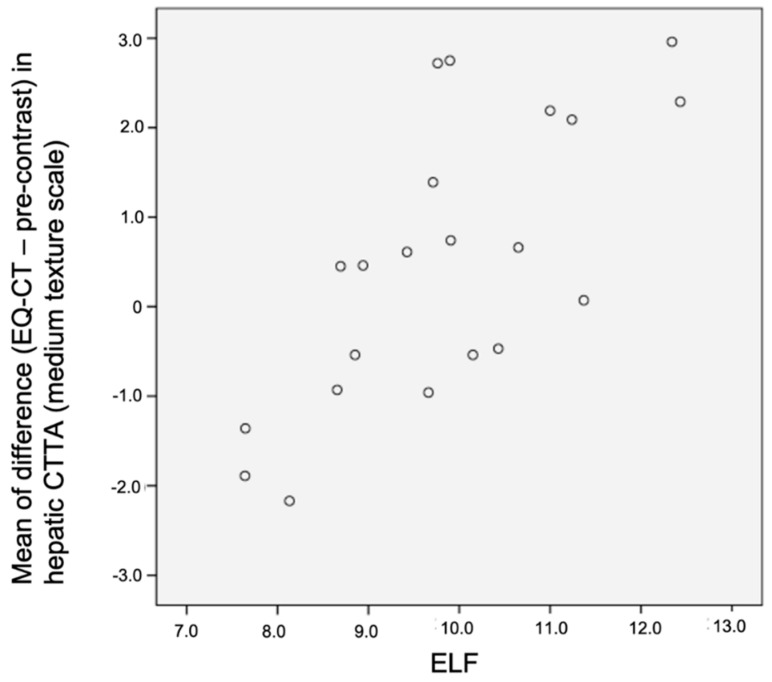
Scatter plot demonstrating correlation between mean of difference in hepatic CTTA versus ELF in segment VII liver texture analysis (r_s_ = 0.69, *p* < 0.001). CTTA: CT texture analysis; ELF: enhanced liver fibrosis panel.

**Figure 6 jpm-10-00046-f006:**
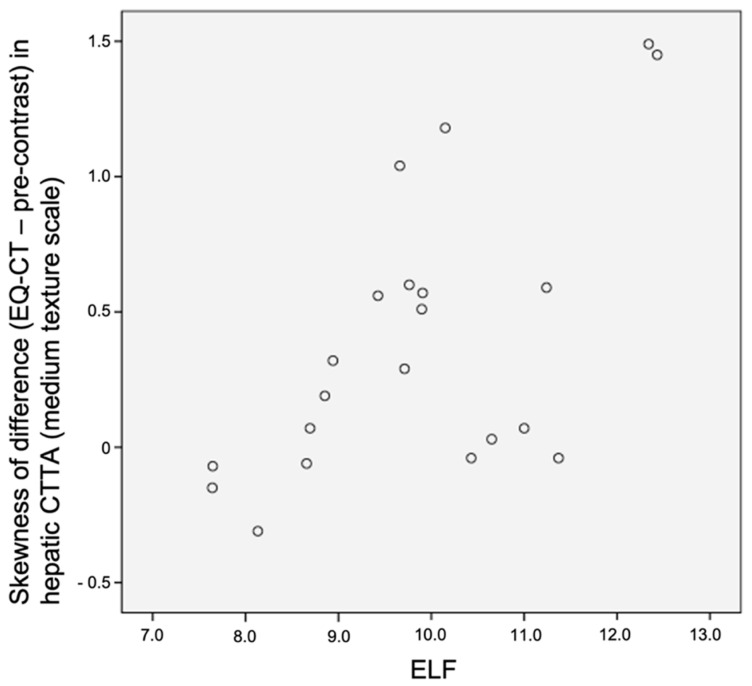
Scatter plot demonstrating correlation between skewness of difference in hepatic CTTA versus ELF in segment VII liver texture analysis (r_s_ = 0.57, *p* = 0.007). CTTA: CT texture analysis; ELF: enhanced liver fibrosis panel.

**Figure 7 jpm-10-00046-f007:**
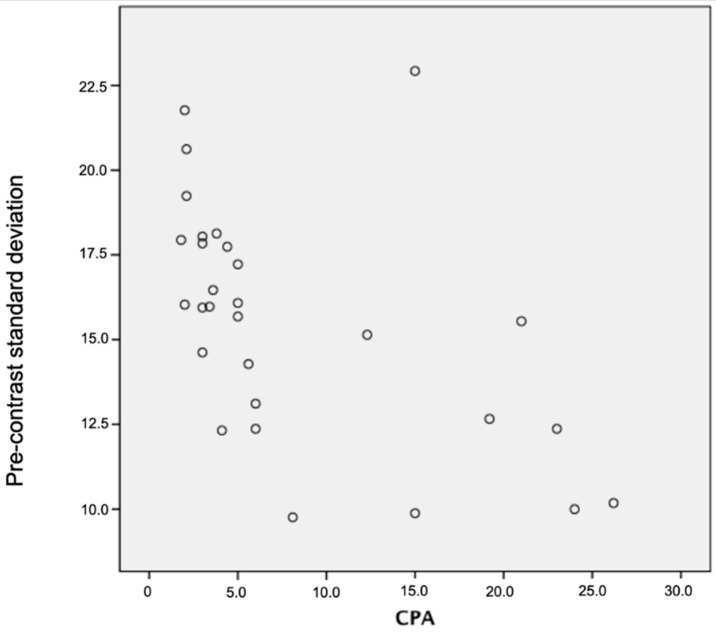
Scatter plot demonstrating correlation between pre-contrast standard deviation versus CPA in whole liver coarse (SSF: 6 mm) texture analysis (r_s_ = −0.66, *p* < 0.001). CPA: collagen-proportionate area; SSF: spatial space filter.

**Figure 8 jpm-10-00046-f008:**
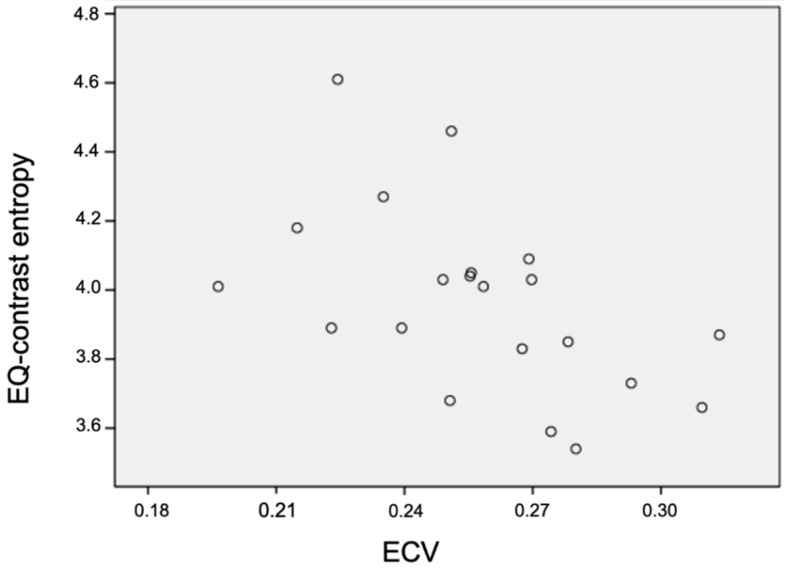
Scatter plot demonstrating correlation between EQ-contrast entropy versus ECV in whole liver coarse (SSF: 6 mm) texture analysis (r_s_ = −0.58, *p* = 0.006). EQ: equilibrium, ECV: extracellular volume, and SSF: spatial scale filter.

**Table 1 jpm-10-00046-t001:** Summary of texture liver analysis results for coarse (SSF: 6 mm) and medium texture scales (SSF: 5, 4 and 3 mm). Bold indicates statistical significance. (r_s_: Spearman’s rank correlation coefficient, CPA: collagen proportionate area, ECV: extracellular volume, SD: standard deviation, MPP: mean positive pixel value, and SSF: spatial scale filter.)

		Whole Liver								Liver Segment VII							
		EQ-contrast minus pre-contrast								EQ-contrast minus pre-contrast							
		Mean				Skewness				Mean				Skewness			
	SSF (mm)	**3**	**4**	5	6	3	4	5	6	3	4	5	6	3	4	5	6
ELF	r_s_	−0.03	−0.088	−0.103	−0.094	−0.352	−0.253	−0.032	0.235	**0.435**	**0.642**	**0.694**	**0.609**	**0.571**	**0.530**	**0.443**	0.282
	*p*-value	0.898	0.703	0.658	0.687	0.118	0.268	0.889	0.305	**0.049**	**0.002**	**<0.001**	**0.003**	**0.007**	**0.013**	**0.044**	0.216

**Table 2 jpm-10-00046-t002:** Texture liver analysis results of mean and skewness at different SSFs vs. ELF. Bold indicates statistical significance. (SSF: spatial scaling factor, r_s_: Spearman’s rank correlation coefficient, ELF: enhanced liver fibrosis, and ECV: extracellular volume.).

			Whole Liver						Liver Segment VII					
			Pre-contrast			EQ-contrast			Pre-contrast			EQ-contrast		
			SD	Entropy	MPP	SD	Entropy	MPP	SD	Entropy	MPP	SD	Entropy	MPP
SSF: 6 mm	CPA	r_s_	**−0.663**	**−0.597**	**−0.554**	**−0.524**	**−0.502**	**−0.584**	**−0.555**	**−0.472**	**−0.368**	−0.404	**−0.601**	−0.378
		*p*-value	**<0.001**	**0.001**	**0.002**	**0.015**	**0.02**	**0.005**	**0.002**	**0.01**	**0.049**	0.07	**0.004**	0.091
	ECV	r_s_	**−0.513**	**−0.463**	**−0.455**	**−0.560**	**−0.582**	**−0.518**	**−0.400**	−0.299	−0.286	**−0.550**	**−0.637**	−0.43
		*p*-value	**0.004**	**0.011**	**0.013**	**0.008**	**0.006**	**0.016**	**0.032**	0.115	0.132	**0.010**	**0.002**	0.052
SSF: 5 mm	CPA	r_s_	**−0.628**	**−0.582**	**−0.585**	**−0.497**	**−0.481**	**−0.520**	**−0.463**	**−0.452**	**−0.386**	−0.364	**−0.574**	**−0.516**
		*p*-value	**<0.001**	**0.001**	**0.001**	**0.022**	**0.027**	**0.016**	**0.012**	**0.014**	**0.038**	0.104	**0.006**	**0.017**
	ECV	r_s_	**−0.483**	**−0.393**	**−0.448**	**−0.571**	**−0.557**	**−0.588**	**−0.368**	−0.318	−0.332	**−0.591**	**−0.628**	−0.414
		*p*-value	**0.008**	**0.035**	**0.015**	**0.007**	**0.009**	**0.005**	**0.05**	0.093	0.078	**0.005**	**0.002**	0.062
SSF: 4 mm	CPA	r_s_	**−0.516**	**−0.479**	**−0.486**	−0.396	−0.386	−0.409	**−0.555**	**−0.472**	**−0.368**	−0.404	**−0.601**	−0.378
		*p*-value	**0.004**	**0.009**	**0.008**	0.076	0.084	0.066	**0.002**	**0.01**	**0.049**	0.07	**0.004**	0.091
	ECV	r_s_	**−0.388**	−0.322	−0.351	**−0.543**	**−0.492**	**−0.555**	**−0.4**	−0.299	−0.286	**−0.550**	**−0.637**	−0.43
		*p*-value	**0.038**	0.088	0.062	**0.011**	**0.023**	**0.009**	**0.032**	0.115	0.132	**0.01**	**0.002**	0.052
SSF: 3 mm	CPA	r_s_	**−0.377**	−0.342	−0.314	−0.383	−0.37	−0.399	**−0.379**	−0.351	−0.332	−0.402	−0.422	−0.427
		*p*-value	**0.044**	0.07	0.097	0.087	0.099	0.073	**0.043**	0.062	0.079	0.071	0.057	0.054
	ECV	r_s_	−0.283	−0.237	−0.214	**−0.449**	−0.381	**−0.487**	−0.347	−0.305	−0.294	**−0.617**	**−0.523**	**−0.591**
		*p*-value	0.137	0.217	0.264	**0.041**	0.088	**0.025**	0.065	0.107	0.122	**0.003**	**0.015**	**0.005**

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
