# Peer review of "Equilibrium CT Texture Analysis for the Evaluation of Hepatic Fibrosis: Preliminary Evaluation against Histopathology and Extracellular Volume Fraction"

_jpm, 2020, doi:10.3390/jpm10020046_

Round 1
Reviewer 1 Report
The Authors of the paper try to evaluate CTTA in pre-contrast and hepatic equilibrium imaging against histologically-quantified fibrosis (CPA), serum-based ELF as well as imaging-based ECV in chronic hepatitis. They indicate that EQ-CTTA can help quantify diffuse fibrosis in chronic liver disease. These results should be validated in a larger group of patients in a prospective study, but the paper is a significant contribution to the scientific discussion about equilibrium CT texture analysis for the evaluation of hepatic fibrosis.
Author Response
Thank you for your review and comments. A future prospective study in a larger cohort will certainly further inform our scientific understanding about the use of CT texture analysis in liver fibrosis.
Reviewer 2 Report
Thank you for a nice article.
Overall
Please check spelling, only minor changes.
Abstract, introduction
Well written and relevant references.
Materials and methods
1. A flowchart figure of inclusion/exclusion would make the presentation more appealing.
2. Page 3 line 110 how come a slice thickness of 10 mm was selected? This could potentially influence texture parameters due to partial volume effect.
Results
In your figures 4, 5 and 7 you only show data from 21 patients and in figure 6 you present all 29. Could you explain this or include all data in the figures.
Discussion
Well discussed results.
Conclusion
Is sound based on the results.
Author Response
Thank you for your comments. I have addressed some of your questions and suggestions as follows:
1) I have included a patient flowchart from the original study as you suggested.
2) Slice thickness of 10 mm was selected in the original study to increase the signal to noise ratio (by averaging over a greater slice thickness). This was a re-analysis of the dataset from the original study hence this could not be altered.
3) Thank you for highlighting the ambiguity with the number of data points within the figures. There were 29 available pre-contrast image datasets available for texture analysis and out of these, 21 EQ-contrast image datasets were available. I have edited and made this point clear in the revised manuscript (i.e. line 74, 77, 92 and 95).